# Effectiveness of A Multifactorial Intervention in Increasing Adherence to the Mediterranean Diet among Patients with Diabetes Mellitus Type 2: A Controlled and Randomized Study (EMID Study)

**DOI:** 10.3390/nu11010162

**Published:** 2019-01-14

**Authors:** Rosario Alonso-Domínguez, Luis García-Ortiz, Maria C. Patino-Alonso, Natalia Sánchez-Aguadero, Manuel A. Gómez-Marcos, José I. Recio-Rodríguez

**Affiliations:** 1Primary Care Research Unit, The Alamedilla Health Center, Biomedical Research Institute of Salamanca (IBSAL), Castilla and León Health Service (SACYL), Department of Nursing and Physiotherapy, University of Salamanca, Spanish Network for Preventive Activities and Health Promotion (redIAPP), 37003 Salamanca, Spain; natalia.san.ag@gmail.com; 2Primary Care Research Unit, The Alamedilla Health Center, Biomedical Research Institute of Salamanca (IBSAL), Castilla and León Health Service (SACYL), Department of Biomedical and Diagnostic Sciences, University of Salamanca, Spanish Network for Preventive Activities and Health Promotion (redIAPP), 37003 Salamanca, Spain; lgarciao@usal.es; 3Biomedical Research Institute of Salamanca (IBSAL), Castilla and León Health Service (SACYL), Department of Statistics, University of Salamanca, Spanish Network for Preventive Activities and Health Promotion (redIAPP), 37003 Salamanca, Spain; carpatino@usal.es; 4Primary Care Research Unit, The Alamedilla Health Center, Biomedical Research Institute of Salamanca (IBSAL), Castilla and León Health Service (SACYL), Department of Medicine, University of Salamanca, Spanish Network for Preventive Activities and Health Promotion (redIAPP), 37003 Salamanca, Spain; magomez@usal.es; 5Biomedical Research Institute of Salamanca (IBSAL), Spanish Network for Preventive Activities and Health Promotion (redIAPP), 37003 Salamanca, Spain; donrecio@gmail.com; 6Faculty of Health Sciences, University of Burgos, 09001 Burgos, Spain

**Keywords:** Mediterranean diet, type 2 diabetes, health education, information and communication technologies

## Abstract

The Mediterranean diet (MD) is recognized as one of the healthiest dietary patterns and has benefits such as improving glycaemic control among patients with type 2 diabetes (T2DM). Our aim is to assess the effectiveness of a multifactorial intervention to improve adherence to the MD, diet quality and biomedical parameters. The EMID study is a randomized and controlled clinical trial with two parallel groups and a 12-month follow-up period. The study included 204 subjects between 25–70 years with T2DM. The participants were randomized into intervention group (IG) and control group (CG). Both groups received brief advice about healthy eating and physical activity. The IG participants additionally took part in a food workshop, five walks and received a smartphone application for three months. The population studied had a mean age of 60.6 years. At the 3-month follow-up visit, there were improvements in adherence to the MD and diet quality of 2.2 and 2.5 points, compared to the baseline visit, respectively, in favour of the IG. This tendency of the improvement was maintained, in favour of the IG, at the 12-month follow-up visit. In conclusion, the multifactorial intervention performed could improve adherence to the MD and diet quality among patients with T2DM.

## 1. Introduction

The term “Mediterranean diet” refers to the eating habits traditionally followed by peoples living around the Mediterranean Sea [1]. It is characterized by the high consumption of vegetables, monounsaturated fatty acids (mainly from olive oil), fruits, whole grains, legumes and fish; moderate consumption of dairy products, fish and red wine; and low consumption of red or processed meats [2,3].

The Mediterranean diet is recognized as one of the healthiest dietary patterns [4]. It is associated with significant improvements in glycaemic control and weight loss [5], as well as with the prevention of cardiovascular diseases, cancer and diabetes mellitus type 2 (T2DM) [6,7,8] due to its antioxidant and anti-inflammatory effects. It has been suggested that the benefits of this pattern T2DM are the result of its effect on insulin sensitivity and resistance, satiety factors and hormone secretion [9], in addition to the high fibre content and low glycaemic index of its components.

Numerous studies have carried out interventions with the aim of increasing adherence to the Mediterranean diet [10,11]. The systematic review by Maderuelo-Fernández et al. [12] showed that the most effective interventions were carried out by trained personnel focused on a population with a disease. More specifically, the interventions carried out by Cubillos et al. [13] and Gomez-Huelgas et al. [14] achieved a two-point increase in adherence to the Mediterranean diet among patients with T2DM. The review by Álvarez-Bueno et al. [15] also showed that multifactorial interventions were the most effective because improvement in lifestyle habits, such as a healthy diet and increased physical activity, are cornerstones in the treatment of T2DM [16].

Kim et al. [17] showed an inverse correlation between dietary improvement and glycaemic control markers. Therefore, diet quality should also be taken into account when assessing the diets of patients with T2DM. Murray et al. [18] concluded that the quality of the diet was lower among this population than the healthy population, even though no differences in the individual intake of macronutrients was detected.

With the advances in new technologies, consideration is needed to determinate how they can best be incorporated as new tools into multifactorial interventions. Mullet et al. [19] concluded that 70% of mHealth interventions were effective in improving diet quality. More specifically, subgroups of the diabetic population that used cell phones obtained greater benefit [20]. Although studies have used mHealth to help improve diet in the diabetic population [21], we have not found interventions that included a smartphone application with the aim of increasing adherence to the Mediterranean diet in this population.

The objective of this study is thus to assess the effectiveness of a multifactorial intervention involving a food workshop and a smartphone application in improving adherence to the Mediterranean diet and diet quality among patients with T2DM.

## 2. Materials and Methods 

### 2.1. Study Design

The EMID study [22] (Effectiveness of a multifactorial intervention based on an application for smartphones, a nutritional workshop and heart-healthy walks, in patients with T2DM in primary care) is a randomized, controlled clinical trial with two parallel groups and a follow-up period of 12 months. The study was carried out in the field of primary health care, in the Alamedilla Research Unit, which belongs to the Research Network on Preventive Activities and Health Promotion (redIAPP) and the Biomedical Research Unit of Salamanca (IBSAL).

### 2.2. Participants

One of the objectives was to obtain a sample that is the most representative of the studied population. Therefore, the subjects were selected by stratified random sampling from among the patients with T2DM who sought medical attention at the Alamedilla Health Centre. The subjects were divided according to their age into two groups (25–50 and 51–70 years).

The inclusion criteria were age between 25 and 70 years, T2DM, agreement to participate in the study and signing an informed consent document after receiving information about the study. As diagnostic criteria for T2DM, we followed the latest recommendations of the American Diabetes Association: fasting plasma glucose above 126 mg/dL or two-hour plasma glucose above 200 mg/dL during an oral glucose tolerance test (using a glucose load containing the equivalent of 75 g anhydrous glucose dissolved in water) or glycosylated haemoglobin over 6.5%. In all cases, these tests were repeated to confirm the results in the absence of unequivocal hyperglycaemia. Additionally, patients were also considered as having T2DM if they had the classic symptoms of hyperglycaemia or hyperglycaemic crisis (i.e., random plasma glucose above 200 mg/dL) [23].

The exclusion criteria were a history of cardiovascular events, musculoskeletal pathology that prevents walking and clinically demonstrable neurological or neuropsychological disease, which would prevent the subject from visiting the health centre.

### 2.3. Common Advice

All participants received standardized counselling for 10 min about healthy eating and physical activity. The food section lasted five minutes and focused on the use of the plate method and recommendations to help comply with the Mediterranean diet. Five minutes were also used to give advice to help comply with current international recommendations regarding physical activity. A brochure was given to the participants for support in both areas.

### 2.4. Randomization and Masking

Participants were randomized at 1:1 into the control group (CG) (102) or the intervention group (IG) (102). The randomization was performed after obtaining informed consent and was not revealed prior to group assignment. The allocation sequence was generated by an independent researcher using the software Epidat 4.0 (Consellería de Sanidade, Santiago de Compostela, Spain) (Figure 1). 

Due to the nature of the intervention, the participants could not be blinded. However, the researcher who carried out the intervention in the study group was different from the person responsible for the assessment and standardized counselling. In addition, the person responsible for the statistical analysis remained blinded throughout the study. During the follow-up visits, patients were told that they should not use other health technologies. Moreover, the application was not made available online until the end of the study so that the control group could not access it.

### 2.5. Intervention

A multifactorial intervention was carried out with groups of 10 participants consisting of a food workshop, a smartphone application and heart-healthy walks. To impart this multifactorial intervention, three nurses from the health centre were instructed in two one-hour training classes on how to carry out the interventions in a standardized manner (what points should be treated, in what order and for how long).

#### 2.5.1. Food Workshop

The food workshop was a theoretical and practical workshop that lasted 90 min and focused on improving adherence to the Mediterranean diet. The workshop covered the following topics: benefits of a healthy diet, food groups, components of the Mediterranean diet, recommended culinary techniques, the use of the plate method and the importance of food labelling for patients with diabetes. Adherence to the food workshop was assessed by the attendance or lack thereof.

#### 2.5.2. Smartphone Application (EVIDENT II)

A one-hour group workshop was held to instruct the participants in the use of the EVIDENT II application (intellectual property registry number SA-81-14) (CGB, Salamanca, Spain), which was installed on a mobile phone that was provided for them to use for three months. This application was designed by software engineers in collaboration with dieticians and physical activity experts with the aim of promoting adherence to the Mediterranean diet and it has already been used in previous studies [24,25]. The application was configured with the data of each participant (age, sex, weight, height and stride distance). After entering food intake and daily exercise data, it would provide detailed information on nutritional deviations in terms of both diet composition and the number of calories, with the aim of encouraging a change of habits. The cell phone was returned after three months, at follow-up visit for both groups. Subsequently, the stored information was downloaded. Adherence to the smartphone app was assessed by the number of days of recordings in the device. After this three-month intervention period, the subjects did not have access to the EVIDENT II app because it was not freely available online. 

#### 2.5.3. Heart-Healthy Walks

Once a week for 5 weeks, the subjects performed 10 min of warm-up, walked 4 km on flat terrain and performed 10 min of stretching and relaxation. In order to make the walks qualify as aerobic exercise (50–70% maximum heart rate) [26], participants were divided into two groups according to on intensity. The approximate speed was 6 km/hour for the group walking at moderate intensity (5 metabolic equivalents (METs)) and 3–4 km/hour for the group walking at low intensity (2.5 METs). Adherence to heart-healthy walks was evaluated by the number of days, planned in the intervention that the subjects attended to perform them.

### 2.6. Outcome Measures and Follow-Up

To assess the effect of the multifactorial intervention, follow-up was carried out at baseline, three months and 12 months after the initial intervention. As the main endpoint, we considered the change in total score of the Mediterranean Diet Adherence Screener (MEDAS) questionnaire [27], while the total score of the Diet Quality Index (DQI) questionnaire [28] and the clinical variables were treated as secondary endpoints.

#### 2.6.1. Adherence to the Mediterranean Diet

The main result variable was measured using the validated fourteen-item MEDAS questionnaire, developed by the PREDIMED group, which includes 12 questions about the frequency of food consumption and two questions about typical eating habits for the Spanish population [27]. Each question was scored with zero or one point. One point was given for using olive oil as the main fat for cooking, preferring white meat to red meat and daily consumption of four or more tablespoons (one tablespoon = 13.5 g) of olive oil (including oil used for frying, dressing salads, etc.), two or more servings of vegetables, three or more pieces of fruit, less than one serving of red meat or sausage, less than one serving of animal fat and less than one cup (one cup = 100 mL) of carbonated or sugary drinks. One point was also given for weekly intake of seven or more glasses of wine, three or more servings of legumes, three or more servings of fish, two shop-bought pastries or fewer, three or more servings of nuts and two or more helpings of sofrito (a traditional sauce made with tomato, garlic, onion or leeks and sautéed with olive oil). The final score range was 0 to 14 points, with 9 or more points indicating suitable adherence to the Mediterranean diet [27].

#### 2.6.2. Diet Quality Index

The diet was assessed with the DQI [28]. This questionnaire covers 18 food groups, labelled one, two or three depending on whether their consumption is beneficial (eating more results in a higher score) or harmful to health (eating more produces a lower score). Food groups are classified into three categories according to the recommended frequency of use. The first category includes eight foods and their response types are “less than once a day,” “once a day” or “more than once a day.” The second category has seven foods and their responses are “less than four times a week,” “four to six times a week” or “at least once a day.” Finally, in the third category, there are three foods, which are classified as “less than twice a week,” “two to three times a week” or “four or more times a week” [28]. Scores range from 18 to 54 points, with a higher score representing higher diet quality and 40 or more points indicate a suitable quality of diet.

#### 2.6.3. Clinically Relevant Measures

Other variables were measured, including drug use, blood pressure, postprandial glucose, weight, height, waist circumference (WC) and biochemical parameters (total serum cholesterol, LDL-cholesterol, HDL-cholesterol). Body mass index (BMI) was calculated by dividing the weight (kg) by the square of the height (m2). The body shape index (ABSI) was estimated with the following equation: ABSI = WC (m)/(BMI2/3 × height (m)1/2). A detailed description of the way in which these variables were measured was published with the study protocol [22].

### 2.7. Sample Size Calculation

The sample size was estimated a priori, considering the increase in the total score of the MEDAS questionnaire as the main endpoint. Assuming alpha = 0.05 and beta = 0.20 with an SD of 2 points, we needed 140 participants (70 per group) in order to detect an increase of 1 point in the IG’s MEDAS score compared to the CG while allowing for an expected drop-out rate of 10%. In addition, for an SD of 2.4 points in the DQI score, we needed 202 participants (101 per group) in order to detect an increase of 1 point in the IG’s DQI score compared to the CG with the same dropout rate. Therefore, we considered 204 subjects to be a sufficient number for detecting clinically relevant differences in the main variables of the study.

### 2.8. Ethical Considerations

The study was approved by the Clinical Research Ethics Committee of the Health Area of Salamanca on 28 November 2016. All procedures were performed in accordance with the ethical standards of the institutional research committee and with the 2013 Declaration of Helsinki [29]. All patients signed written informed consent documents prior to participation in the study. The trial was registered at ClinicalTrials.gov with identifier NCT02991079.

### 2.9. Statistical Analysis

Continuous variables are presented as means ± standard deviations and qualitative variables are presented using a frequency distribution. To compare categorical variables at baseline between IG and CG the chi-squared test or Fisher exact test was carried out as appropriate and quantitative variables were compared using the student’s *t*-test. The ANCOVA test was used to compare the changes between the IG and the CG, adjusting for the basal measurement of each variable. Repeated-measures analysis of variance was used to analyse group interaction effects on changes in the Mediterranean diet and DQI scores using the General linear Model (GLM) procedure. For the bilateral contrast of hypotheses, an alpha risk of 0.05 was set as a limit of statistical significance. The data were analysed using the statistical software SPSS for Windows version 25.0. (IBM Corp, Armonk, NY, USA).

## 3. Results

### 3.1. Sample Selection

Among the 1291 subjects with T2DM between 25–70 years old seen at the La Alamedilla Health Centre, 473 were selected by stratified random sampling. Subsequently, 103 were excluded because they did not meet the inclusion criteria, 137 did not want to be included in the study and 29 were excluded for other reasons. Finally, 204 subjects were included in the study. There were 19 subjects who were lost in follow up (11 in the CG and 8 in the IG, *p* > 0.05). The reasons for dropping out are detailed in the flowchart in Figure 1.

### 3.2. Demographic and Clinical Characteristics of the Included Subjects

Table 1 presents the baseline characteristics of the participants. The study population had an average age of 60.6 ± 8.1 years and 45.6% (*N* = 93) were women. At the baseline visit, there were no significant differences between the groups in terms of demographic characteristics.

### 3.3. Adherence to the Intervention

The adherence to the food workshop was 82.3% (84/102 subjects). The majority of the subjects (63.7%, 65/102 subjects) carried out between 4 to 5 heart-healthy walks. The average use of the application was 35 days and most of the subjects (69.6%, 71/102) used it for 31 to 60 days.

### 3.4. Effect of the Intervention

In the CG, the scores of the adherence to the Mediterranean diet showed few variations throughout the study (6.9, 7.1 and 7.0 points, respectively) and the same thing happened in the quality of the diet (40.2, 40.8 and 40.4 points, respectively). However, in the IG, an increase in adherence to the Mediterranean diet was observed in the follow-up visits with respect to the baseline visit (7.2, 9.4 and 8.5 points, respectively), as well as in the quality of the diet (39.8, 42.9 and 41.8 points, respectively), as shown in Table 2.

Table 3 shows the intergroup modifications that occurred in the follow-up visits with respect to the baseline visit. At the 3-month follow-up visit, compared to the baseline visit, there was an improvement in the adherence to the Mediterranean diet and the quality of the diet of 2.2 points (1.8–2.5) and 2.5 points (1.9–3.0), respectively, in favour of the IG. In the same way, these improvements were maintained in the follow-up visit at 12 months, with an increase of 1.3 points (0.8–1.8) in the adherence to the Mediterranean diet and 1.7 points (1.0–2.4) in the quality of the diet, with respect to the baseline visit (*p* < 0.01 for all). At the 3-month follow-up visit, 82.7% of the IG participants increased their MEDAS questionnaire score by at least 1 point, while in the CG this value was 31.3%; at the 12-month follow-up visit, 61.9% of the IG participants and 38.5% of the CG participants reached this increase.

Taking into account the terms of the MEDAS questionnaire, there was an improvement in the follow-up visit at 3 months, in favour of the IG, in the daily use of at least 4 tablespoons of olive oil, 2 servings or more vegetables and at least 3 servings of fruits, as well as in the weekly consumption of at least 3 servings of fish or seafood, 2 or fewer servings of commercial baked goods, at least 3 servings of nuts, 2 or more servings of sofrito sauce and finally and a greater consumption of white meats than red meats. The improvements were maintained at 12 months of follow-up, in the consumption of olive oil, fish or seafood, nuts, white meats and sofrito sauce. (*p* < 0.05 for all; Table 3).

Figure 2 shows the evolution of the percentage of compliance of the Mediterranean diet and the DQI index in the IG and CG. At the follow-up visits at 3 and 12 months, the number of subjects with suitable adherence to the Mediterranean diet (MEDAS score ≥ 9 points) in the IG increased significantly compared to the CG. In the same way, there was an increase in the IG of the number of subjects with a suitable quality of diet (DQI score ≥ 40 points), with respect to the CG. This difference was significant in the follow-up visit at 3 months. (*p* < 0.05 for all).

In addition, a significant interaction effect (*p* < 0.001) was found between the IG and the changes in adherence to the Mediterranean diet and diet quality throughout the 12 months of follow-up. (Figure 3).

Finally, without significant changes in the consumption of antidiabetic drugs, the following improvements were observed at the 3-month follow-up visit compared to the baseline visit: a postprandial glucose improvement of −9.4 mg/dL (−18.0–−0.8), a waist circumference improvement of −2.5 cm (−3.3–−1.6) and a BMI improvement of −0.3 kg/m^2^ (−0.6–0.0) in favour of the IG (*p* < 0.05 for all; Table 4 and Table 5). The improvements of some of the clinical parameters were maintained at the follow-up visit at 12 months compared to baseline, such as a postprandial glucose improvement of −6.6 mg/dL (−14.9–1.7), waist circumference improvement of −0.9 cm (−3.1–1.2) and systolic blood pressure improvement of −1.5 mmHg (−8.7–5.7) However, the results did not reach statistical significance (*p* > 0.05).

## 4. Discussion

The main findings of the study were that a multifactorial intervention involving a food workshop and a smartphone application produced improved adherence to the Mediterranean diet and diet quality among patients with T2DM in comparison to standard treatment.

In line with our results, Zazpe et al. [30] obtained significant increases of 1.4 and 1.8 points in Mediterranean diet adherence among patients with T2DM or cardiovascular risk factors in their intervention groups, who were instructed to adhere to a Mediterranean diet supplemented with virgin olive oil or mixed nuts. This is an important finding since several studies have concluded that greater adherence to the Mediterranean diet results in an improvement in glycaemic control [31], insulin sensitivity [32] and insulin secretion [33], which are important factors in the management of T2DM.

Similarly, when breaking down the adherence to the Mediterranean diet by item, the results are in line with those published by Salas-Salvado et al. [10] and Estruch et al. [34]. In addition, in this study, the preference for white meat over red meat and the consumption of homemade sofrito at least twice a week were modified. Several studies have explained the clinical implication that the increases in Mediterranean diet items could have independently. Guasch-Ferré et al. [35], linked the increase in the consumption of 10 g per day of olive oil to a 16% reduction in cardiovascular risk. Furthermore, according to Salas-Salvado et al. [10], the consumption of nuts has a beneficial effect on insulin resistance, blood pressure, dyslipidaemia and functions that influence the modulation of inflammation and endothelial function. Moreover, eating fish [36] and sofrito [37] is related to beneficial effects on cardiovascular risk factors. Kim et al. [38], linked a reduction in the consumption of red meats to an improvement in insulin sensitivity.

These changes in dietary habits resulted in changes in the clinical variables. These changes correspond to those achieved by other multifactorial interventions such as that of Jiang et al. [39], who observed a decrease in postprandial glycaemia (10.7 ± 3.2 vs. 5.8 ± 2.1 mmol/L) in the IG after 12 months of follow-up, through changes in diet, exercise and a psychological approach. In the same way, several studies [39,40,41,42] have managed to significantly improve BMI among other parameters through a multifactorial intervention based on changes in diet and exercise in subjects with T2DM. Finally, we think that the results obtained in this work could be pioneering, since as far as we know, this study is the first to analyse the effects of a multifactorial intervention on diet in subjects with T2DM based on a food workshop and a smartphone application, as well as achieving improvement in the MEDAS and the DQI questionnaire.

This study has several limitations that should be considered when interpreting our results. First, the exposure to the intervention was short (three months). In addition, the multifactorial nature of the intervention means that it is not possible to know which component produced the change in the study group. Furthermore, the main findings of the study are based on patients’ responses on nutrition questionnaires. Finally, although the recommendation was made not to use other mobile applications, we have no guarantee that the participants did not do so.

## 5. Conclusions

The results of this study suggest that the proposed multifactorial intervention involving a food workshop and a smartphone application are moderately effective in improving adherence to the Mediterranean diet and diet quality among patients with T2DM.

## Figures and Tables

**Figure 1 nutrients-11-00162-f001:**
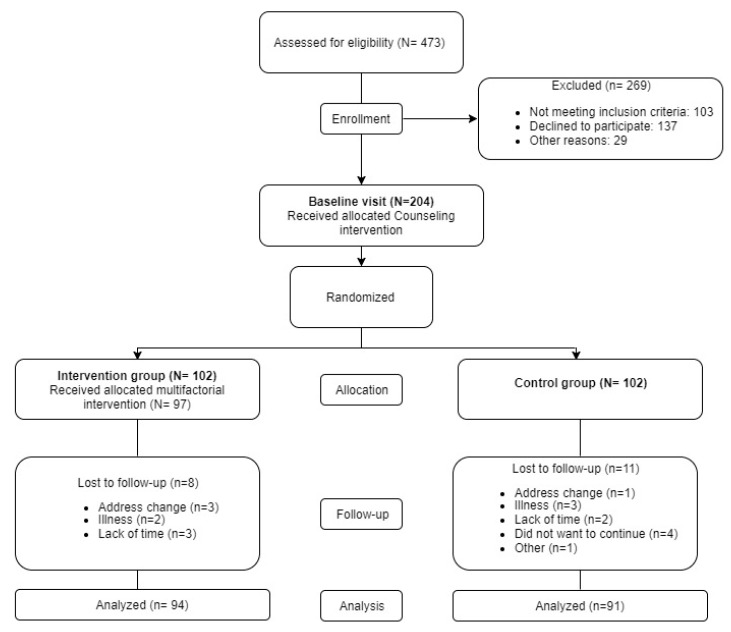
Flow-chart of the EMID Study: enrolment of the participants and completion of the study.

**Figure 2 nutrients-11-00162-f002:**
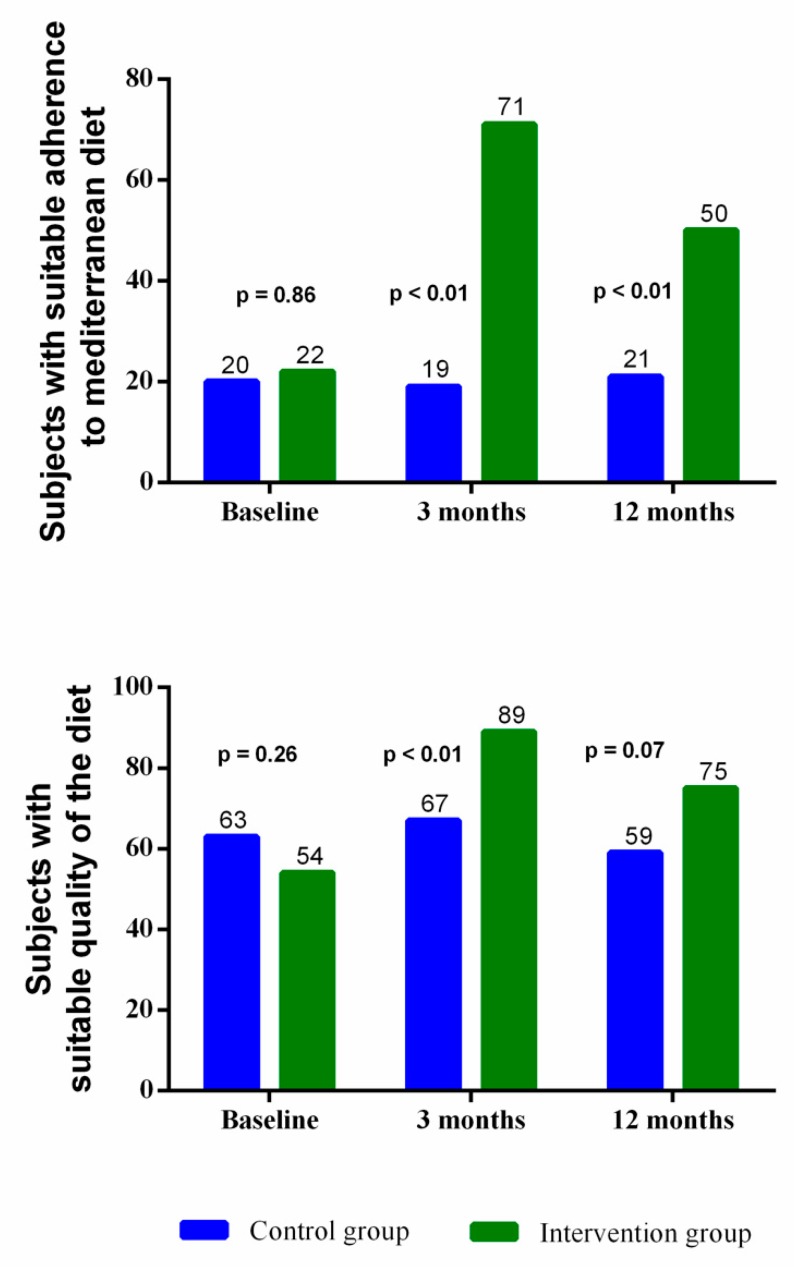
Changes in the number of subjects with suitable adherence to the Mediterranean diet and quality of diet. The chi-squared test or Fisher exact test was carried out.

**Figure 3 nutrients-11-00162-f003:**
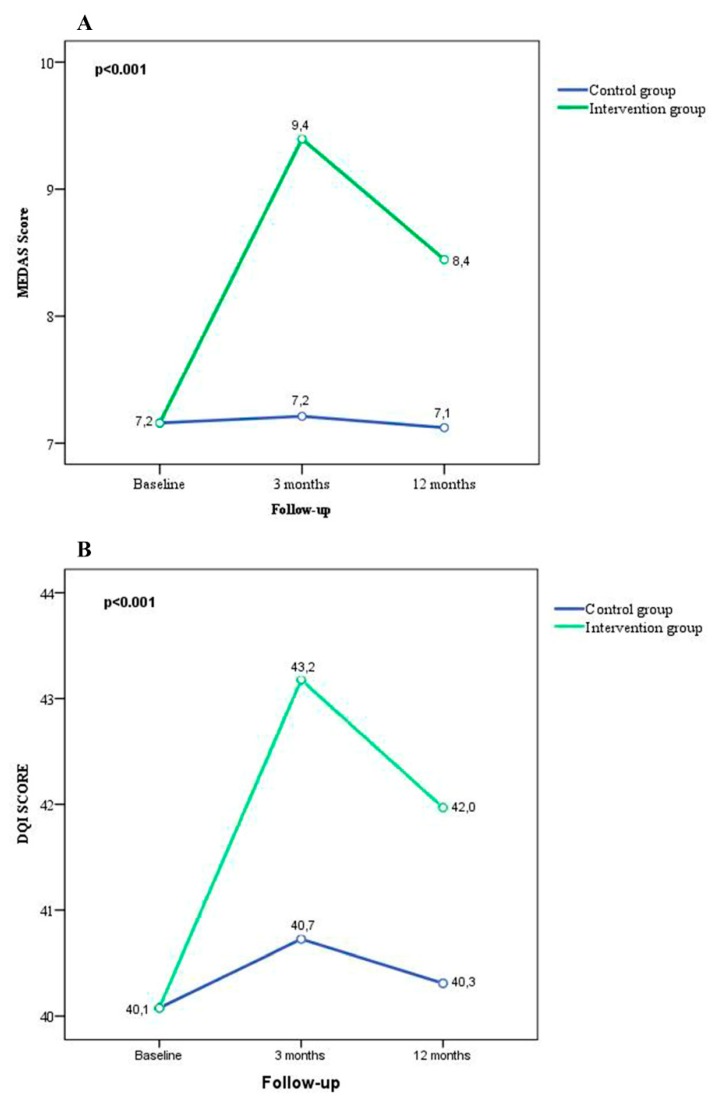
Changes in the MEDAS score and the DQI score tasks in response to the multifactorial intervention. Repeated-measures ANOVA was been performed with adjustment for the baseline visit. MEDAS: Mediterranean Diet Screener; DQI: Diet Quality Index.

**Table 1 nutrients-11-00162-t001:** Baseline characteristics of patients.

	Control Group (102)	Intervention Group (102)	*p*-Value
**Age (years), mean (SD)**	60.4 (8.4)	60.8 (7.8)	0.84
**Gender (female), *n* (%)**	41 (40.2)	52 (51.1)	0.08
**Diabetes duration (years), mean (SD)**	6.7 (4.7)	6.4 (4.6)	0.62
**BMI (kg/m^2^), mean (SD)**	30.3 (5.6)	29.5 (4.2)	0.27
**Obesity, *n* (%)**	48 (47.1)	43 (42.2)	0.48
**Work situation, *n* (%)**			
Works outside home	41 (40.2)	29 (28.4)	0.91
Homemaker	18 (17.6)	24 (23.5)	
Retired	34 (33.3)	44 (43.1)	
Unemployed	7 (6.8)	5 (4.9)	
**Educational level, *n* (%)**			
University studies	17 (16.7)	17 (16.7)	0.94
Middle or high school	33 (32.4)	37 (36.3)	
Elementary school	52 (51.0)	48 (47.1)	
**Smoking, *n* (%)**			
Non-smoker	37 (36.3)	39 (38.2)	0.13
Smoker	26 (25.5)	10 (9.8)	
Former smoker	39 (38.2)	53 (51.9)	
**Medication use, *n* (%)**			
Antihypertensive drugs	55 (53.9)	58 (56.9)	0.24
Lipid-lowering drugs	59 (57.8)	58 (52.0)	0.39
Antidiabetic drugs	89 (87.3)	93.0 (91.2)	0.37
Insulins	18 (17.6)	12 (11.8)	0.24
Metformin	81 (79.4)	82 (80.4)	0.86
Sulphonylureas	8 (7.8)	6 (5.9)	0.58
Thiazolidinediones	3 (2.9)	3 (2.9)	1.0
SGLT2 Inhibitors	1 (1.0)	2 (2.0)	0.56
Meglitinides	2 (2.0)	4 (3.9)	0.41
DPP-4 inhibitors	38 (37.3)	37 (36.3)	0.89
GLP-1 receptor agonists	1 (1.0)	4 (3.9)	0.18

Variables are given as means ± standard deviations, numbers and %. *p* value differences between control group and intervention group. Abbreviations: IQR: Interquartile Range; SD: Standard Deviation; BMI: Body Mass Index; SGLT2: Sodium-glucose co-transporter-2; DPP-4: Dipeptidyl peptidase 4; GLP-1: Glucagon-like peptide-1.

**Table 2 nutrients-11-00162-t002:** Adherence to the Mediterranean diet and diet quality throughout the study.

	Baseline		3-Month Follow Up		12-Month Follow Up	
	Control Group	Intervention Group	Control Group	Intervention Group	Control Group	Intervention Group
	Mean (SD)	Mean (SD)	Mean (SD)	Mean (SD)	Mean (SD)	Mean (SD)
**Total score (points)**						
Mediterranean Diet Adherence Screener	6.9 (1.7)	7.2 (1.9)	7.1 (1.7)	9.4 (1.7)	7.0 (1.9)	8.5 (1.9)
Diet Quality Index	40.2 (2.6)	39.8 (2.6)	40.8 (2.7)	42.9 (2.5)	40.4 (2.7)	41.8 (2.8)
**Mediterranean diet criteria (%)**						
Use olive oil as main culinary fat	89.0 (31.2)	96.0 (19.5)	92.0 (27.8)	99.0 (10.1)	88.0 (32.8)	98.0 (14.3)
Olive oil ≥ 4 tablespoons	15.0 (35.6)	17.0 (37.5)	17.0 (37.5)	43.0 (49.7)	20.0 (40.1)	48.0 (50.2)
Vegetables ≥ 2 servings/day	20.0 (39.9)	24.0 (42.6)	23.0 (42.3)	52.0 (50.2)	22.0 (41.6)	34.0 (47.6)
Fruits ≥ 3 servings/day	51.0 (50.2)	54.0 (50.1)	59.0 (49.4)	76.0 (43.2)	57.0 (49.8)	64.0 (48.3)
Red or processed meats < 1 serving/day	97.0 (17.0)	91.0 (28.5)	96.0 (20.1)	93.0 (25.9)	91.0 (28.5)	92.0 (27.7)
Butter, cream or margarine <1 serving/day	92.0 (27.0)	86.0 (34.6)	93.0 (26.1)	93.0 (26.7)	95.0 (22.9)	91.5 (26.4)
Sugar-sweetened beverage < 1 cup/day	76.0 (42.6)	88.0 (32.4)	87.0 (33.2)	97.0 (17.3)	86.0 (35.2)	93.0 (25.1)
Red wine ≥ 7 servings/week	15.0 (35.6)	19.0 (39.1)	12.0 (33.2)	20.0 (40.5)	14.0 (35.2)	15.0 (36.3)
Legumes ≥ 3 servings/week	19.0 (39.1)	21.0 (40.6)	26.0 (44.1)	34.0 (47.5)	25.0 (43.7)	26.0 (44.0)
Fish or seafood ≥ 3 servings/week	42.0 (49.6)	51.0 (50.2)	47.0 (50.2)	66.0 (47.5)	44.0 (49.9)	61.0 (49.1)
Commercial bakery ≤ 2 servings/week	47.0 (50.2)	32.0 (47.0)	38.0 (48.7)	46.0 (50.1)	31.0 (46.4)	29.0 (45.5)
Nuts ≥ 3 servings/week	24.0 (42.6)	27.0 (44.8)	28.0 (45.2)	51.0 (50.2)	20.0 (40.1)	42.0 (49.7)
White meats more than red meats	58.0 (49.6)	71.0 (45.8)	56.0 (49.9)	89.0 (31.7)	66.0 (47.7)	87.0 (34.2)
Use of sofrito sauce ≥ 2 servings/week	49.0 (50.2)	48.0 (50.0)	39.0 (48.9)	83.0 (38.1)	45.0 (50.0)	66.0 (47.6)

**Table 3 nutrients-11-00162-t003:** Intergroup modifications of the diet parameters in the follow-up visits compared to the baseline visit.

	3 Months vs. Baseline		12 Months vs. Baseline	
	Mean (95% CI)	*p*	Mean (95% CI)	*p*
**Total score (points)**				
Mediterranean Diet Adherence Screener	2.2 (1.8–2.5)	<0.001	1.3 (0.8–1.8)	<0.001
Diet Quality Index	2.5 (1.9–3.0)	<0.001	1.7 (1.0–2.4)	<0.001
**Mediterranean diet criteria (%)**				
Use olive oil as main culinary fat	4.3 (−5.0–9.1)	0.080	0.8 (1.4–14.5)	0.017
Olive oil ≥ 4 tablespoons	25.9 (14.3–37.5)	<0.001	28.4 (15.5–41.4)	<0.001
Vegetables ≥ 2 servings/day	26.8 (16.1–37.5)	<0.001	10.8 (1.6–23.2)	0.087
Fruits ≥ 3 servings/day	15.0 (4.8–25.2)	0.004	4.9 (−7.6–17.3)	0.441
Red or processed meats < 1 serving/day	3.4 (−10.0–3.2)	0.313	1.1 (−7.1–9.2)	0.793
Butter, cream or margarine < 1 serving/day	3.7 (−2.1–9.6)	0.211	3.2 (−1.7–8.9)	0.325
Sugar-sweetened beverage < 1 cup/day	5.4 (−1.2–11.9)	0.109	2.2 (−6.2–10.5)	0.608
Red wine ≥ 7 servings/week	5.3 (−2.3–12.8)	0.169	−0.3 (−7.6–7.1)	0.945
Legumes ≥ 3 servings/week	6.6 (−5.7–18.8)	0.292	0.6 (−11.4–12.5)	0.926
Fish or seafood ≥ 3 servings/week	12.8 (2.3–23.4)	0.018	12.9 (0.2–25.6)	0.046
Commercial bakery ≤ 2 servings/week	18.1 (6.1–30.0)	0.003	3.9 (−8.5–16.2)	0.537
Nuts ≥ 3 servings/week	22.1 (10.1–34.1)	<0.001	21.9 (9.9–33.9)	<0.001
White meats more than red meats	27.3 (17.3–37.3)	<0.001	16.1 (5.4–26.9)	0.004
Use of sofrito sauce ≥ 2 servings/week	44.1 (31.8–56.4)	<0.001	21.0 (7.0–35.0)	0.004

*p* value differences between intervention group and control group. Significant difference: *p* < 0.05. An analysis of covariance adjusting for baseline was used to examine the significance of differences.

**Table 4 nutrients-11-00162-t004:** Clinical variables throughout the study.

	Baseline		3-Month Follow Up		12-Month Follow Up	
	Control Group	Intervention Group	Control Group	Intervention Group	Control Group	Intervention Group
	Mean (SD)	Mean (SD)	Mean (SD)	Mean (SD)	Mean (SD)	Mean (SD)
Glycated haemoglobin (%)	6.8 (1.2)	6.9 (1.2)	6.8 (1.1)	6.8 (1.0)	7.0 (1.3)	6.9 (1.1)
Postprandial glucose (mg/dL)	147.6 (35.5)	149.2 (39.0)	148.9 (37.8)	140.0 (31.7)	148.3 (34.1)	143.2 (29.0)
Atherogenic index	3.8 (1.1)	3.5 (1.0)	3.7 (1.1)	3.4 (1.0)	3.6 (1.1)	3.4 (1.1)
Total serum cholesterol (mg/dL)	176.4 (31.7)	178.8 (30.3)	173.1 (29.0)	173.7 (30.6)	173.0 (32.5)	174.8 (34.0)
LDL-cholesterol (mg/dL)	100.4 (28.6)	101.8 (30.0)	97.9 (24.9)	97.2 (27.9)	94.9 (25.3)	95.6 (31.6)
HDL-cholesterol (mg/dL)	50.3 (14.8)	54.4 (14.1)	49.8 (13.6)	54.0 (14.6)	51.4 (13.5)	54.1 (14.8)
Waist circumference (cm)	104.9 (13.1)	102.2 (11.5)	105.5 (13.0)	100.4 (11.0)	104.2 (16.0)	101.2 (11.8)
Systolic blood pressure (mmHg)	135.0 (33.2)	133.2 (15.9)	131.2 (16.4)	127.1 (15.9)	130.3 (16.4)	126.2 (15.8)
Diastolic blood pressure (mmHg)	80.5 (9.6)	80.8 (9.0)	79.7 (9.6)	78.8 (9.5)	80.6 (8.8)	79.4 (8.5)
ABSI × 100	8.5 (0.5)	8.5 (0.4)	8.5 (0.6)	8.3 (0.4)	8.5 (0.7)	8.4 (0.4)

Abbreviations: LDL: Low density lipoprotein; HDL: High density lipoprotein; ABSI: A Body Shape Index.

**Table 5 nutrients-11-00162-t005:** Intergroup modifications of the clinical parameters in the follow-up visits compared to the baseline visit.

	3 Months vs. Baseline		12 Months vs. Baseline	
	Mean (95% CI)	*p*	Mean (95% CI)	*p*
Glycated haemoglobin (%)	−0.1 (−0.3–0.0)	0.145	−0.1 (−0.4–0.1)	0.241
Postprandial glucose (mg/dL)	−9.4 (−18.0–−0.8)	0.031	−6.6 (−14.9–1.7)	0.118
Atherogenic index	−0.1 (−0.3–0.1)	0.324	0.0 (−0.2–0.2)	0.998
Total serum cholesterol (mg/dL)	−1.8 (−8.1–4.5)	0.567	−0.9 (−8.5–6.8)	0.823
LDL-cholesterol (mg/dL)	−2.6 (−8.3–3.2)	0.380	−0.1 (−6.5–6.3)	0.973
HDL-cholesterol (mg/dL)	0.8 (−1.5–3.1)	0.501	−0.1 (−2.4–2.2)	0.927
Waist circumference (cm)	−2.5 (−3.3–−1.6)	<0.001	−0.9 (−3.1–1.2)	0.390
Systolic blood pressure (mmHg)	−3.1 (−7.1–0.8)	0.120	−1.5 (−8.7–5.7)	0.678
Diastolic blood pressure (mmHg)	−1.2 (−3.3–0.9)	0.244	−1.3 (−3.5–0.8)	0.227
ABSI × 100	0.1 (−0.2–−0.1)	0.001	0.1 (−0.2–0.0)	0.198

An analysis of covariance adjusting for baseline was used to examine the significance of differences. *p* value differences between intervention group and control group. Significant difference: *p* < 0.05. Abbreviations: CI: Confidence Interval; LDL: Low density lipoprotein; HDL: High density lipoprotein; ABSI: A Body Shape Index.

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
