# Peer review of "Effectiveness of A Multifactorial Intervention in Increasing Adherence to the Mediterranean Diet among Patients with Diabetes Mellitus Type 2: A Controlled and Randomized Study (EMID Study)"

_nutrients, 2019, doi:10.3390/nu11010162_

Round 1

Reviewer 1 Report

General comments

The study evaluates the effectiveness of a multifactorial intervention in increasing adherence   to the  Mediterranean diet in people with type 2 diabetes. The study results  show a very small (if any) clinical impact and in my opinion the study  conclusions are not fully supported by the results. Other relevant methodological issues are dealt with below.

Specific commnets

Study design participants were  sampled within age strata (25-35,36-50 and 51-70 years). Type 2 diabetes   in the age range 25-35 year is very rare, the authors should explain what purpose served this  sampling technique and how many  patients  were recruited within each age stratum.

The criteria for the definition of diabetes include  OGTT, how many were  diagnosed  by  OGTT? Was this number balanced between the intervention and control group?

Further details on treatment for  diabetes  in the two study arms should be provided; apparently  metformin and insulin were the only drugs used (table 1) which is surprising  in view of the plethora of available agents for the treatment of type 2 diabetes.

Interventions

the interventions included  among others  “hearth-healthy  walks” , in what did they consists  ( i.e.  intensity , frequency etc.. ),  how was adherence  assessed? How does this intervention fit  with the improvement in adherence to Med Diet?

Outcomes   the authors should indicate the magnitude of the change they want to assess  with regard to  adherence to med diet and diet quality index. The sample size was calculated based on the detection of one point increase in the adherence score suggesting that this is the principal outcome , but in the  analyses the scores are treated as  continuous variables.

 Results 

table 1  should include  BMI, proportion of  obese participants and diabetes duration

Figure 2   the analysis reported in this figure should be better described in the text. The meaning of   “suitable adherence to the Mediterranean diet and quality of diet” should be clarified; furthermore  these measures  were not described  among the study end points

 Figure 3 are these numbers  the same reported in table  2 (first two lines)? If  not , please explain the difference? If yes, please explain why a statistical significance is observed in the figure  but not in the table.

Data on clinical variables  are relevant as the final aim of improving  adherence to diet is  to improve  the management of diabetes, they should be  given in the paper  and not reported as appendix tables. With regard to these data it is important to clarify whether changes in the use of drugs occurred  between baseline and follow-up.

Author Response

REVIEWER #1:

General comments

The study evaluates the effectiveness of a multifactorial intervention in increasing adherence  to the  Mediterranean diet in people with type 2 diabetes. The study results  show a very small (if any) clinical impact and in my opinion the study conclusions are not fully supported by the results. Other relevant methodological issues are dealt with below.

Authors' Answer

First, thank you for your work in reviewing this manuscript. Your contributions and suggestions will improve the understanding of the text.

Following the recommendations of the reviewer, we have modified our conclusions:

Page 16 Line 350: The results of this study suggest that the proposed multifactorial intervention involving a food workshop and a smartphone application, could be effective in improving adherence to the Mediterranean diet and diet quality among patients with T2DM.

Specific commnets

1.      Study design

1.1.Participants were  sampled within age strata (25-35, 36-50 and 51-70 years). Type 2 diabetes in the age range 25-35 year is very rare, the authors should explain what purpose served this sampling technique and how many  patients  were recruited within each age stratum.

Authors' Answer

We agree with the reviewer that Type 2 Diabetes in the age range 25-35 years is very rare, but when we conducted the search for the population susceptible to be studied, we observed that there were subjects with Type 2 Diabetes in this range. However, to improve the understanding of the manuscript, we decided to merge the group between 25-35 years and 36-50 years, forming a group between 25-50 years, being in the manuscript as follows:

Page 3 Line 94: One of the objectives was to obtain a sample that is the most representative of the studied population. Therefore, the subjects were selected by stratified random sampling from among the patients with T2DM who sought medical attention at the Alamedilla Health Centre. The subjects were divided according to their age into two groups (25-50 and 51-70 years).

For the sample to be the most representative of the reference population, the sampling was carried out by stratified random sampling by age, respecting the proportion of subjects of each group, susceptible to be selected for the study. Among the 1291 subjects with T2DM between 25 and 70 years old: 109 belonged to the group between 25-50 years (8.5% of the total population), including 19 subjects in this age range (9.3% of the participating subjects); on the other hand, 1181 subjects belonged to the group between 51-70 years (91.5% of the total population), being included in the study 185 subjects of this group (90.7% of the participating subjects).

1.2.The criteria for the definition of diabetes include OGTT, how many were  diagnosed  by  OGTT? Was this number balanced between the intervention and control group?

Authors' Answer

In our population, 10 subjects were diagnosed of diabetes by OGTT. There is a balance in this number between the intervention group and control group, since 5 subjects were diagnosed in each group by OGTT.

1.3.Further details on treatment for  diabetes  in the two study arms should be provided; apparently  metformin and insulin were the only drugs used (table 1) which is surprising  in view of the plethora of available agents for the treatment of type 2 diabetes.

Authors' Answer

Following the recommendations of the reviewer we have included, in more detail, the antidiabetic drugs in Table 1.

2.      Interventions

a) The interventions included among others “hearth-healthy walks”, in what did they consists ( i.e.  intensity, frequency etc.. )

Authors' Answer

We have included more information about heart-healthy walks in the methods and materials section:

Page 5 Line 155: 2.5.3. Heart-healthy walks

Once a week for 5 weeks, the subjects performed 10 minutes of warm-up, walked 4 km on flat terrain, and performed 10 minutes of stretching and relaxation. In order to make the walks qualify as aerobic exercise (50 - 70% maximum heart rate) [26], participants were divided into two groups according to on intensity. The approximate speed was 6 km/hour for the group walking at moderate intensity (5 metabolic equivalents [METs]) and 3 - 4 km/hour for the group walking at low intensity (2.5 METs). Adherence to heart-healthy walks was evaluated by the number of days, planned in the intervention that the subjects attended to perform them.

b)     How was adherence assessed?

Authors' Answer

We have included an explanation for this in the methods section:

Page 4 Line 140: Adherence to the food workshop was assessed by the attendance or lack thereof.

Page 5 Line 151: Adherence to the smartphone app was assessed by the number of days of recordings in the device.

Page 5 Line 161: Adherence to heart-healthy walks was evaluated by the number of days, planned in the intervention that the subjects attended to perform them.

In addition, we have included a paragraph in the results section about adherence to the intervention:

Page 7 Line 244: 3.3. Adherence to the intervention

The adherence to the food workshop was 82.3% (84/102 subjects). The majority of the subjects (63.7%, 65/102 subjects) carried out between 4 to 5 heart-healthy walks. The average use of the application was 35 days, and most of the subjects (69.6%, 71/102) used it for 31 to 60 days.

c)      How does this intervention fit with the improvement in adherence to Med Diet?

Authors' Answer

We understand the doubts of the reviewer. When carrying out the bibliographic search, in order to propose our study, we found that the most effective interventions were multifactorial interventions. Since diet and exercise are the cornerstones in the control and follow-up of Type 2 diabetes Mellitus, we decided to test in our study an intervention (consisting of heart-healthy walks, a food workshop and an application for Smartphone), with the objectives of improving adherence to the Mediterranean diet and glycemic control, and increasing physical activity.

In this manuscript, we present the efficacy of the intervention in the improvement of adherence to the Mediterranean diet and clinical variables.

However, the efficacy of the intervention in the increase of physical activity has been analyzed as an objective in another manuscript.

3.      Outcomes  

3.1.a) The authors should indicate the magnitude of the change they want to assess  with regard to  adherence to med diet and diet quality index.

Authors' Answer

Following the recommendations of the reviewer, we have included the magnitude of the change that we want to evaluate with respect to diet quality index, and rewritten the paragraph to improve the understanding:

Page 6 Line 202: The sample size was estimated a priori, considering the increase in the total score of the MEDAS questionnaire as the main endpoint. Assuming alpha=0.05 and beta=0.20 with an SD of 2 points, we needed 140 participants (70 per group) in order to detect an increase of 1 point in the IG's MEDAS score compared to the CG while allowing for an expected drop-out rate of 10%. In addition, for an SD of 2.4 points in the DQI score, we needed 202 participants (101 per group) in order to detect an increase of 1 point in the IG’s DQI score compared to the CG with the same dropout rate. Therefore, we considered 204 subjects to be a sufficient number for detecting clinically relevant differences in the main variables of the study.

b) The sample size was calculated based on the detection of one point increase in the adherence score suggesting that this is the principal outcome, but in the  analyses the scores are treated as  continuous variables.

Authors' Answer

The size of the sample was calculated based on the main result of this study, which is the increase in the total score of the MEDAS questionnaire, measured quantitatively.

To avoid confusion with the Figure 2, in which adherence to the Mediterranean diet is used as a dichotomous variable, (suitable adherence: MEDAS ≥ 9 points; non suitable adherence: MEDAS < 9 points) we have introduced a new sentence referring to the endpoint of the study in the materials and methods section:

Page 5 Line 165: As the main endpoint, we considered the change in total score of the Mediterranean Diet Adherence Screener (MEDAS) questionnaire [27], while the total score of the Diet Quality Index (DQI) questionnaire [28] and the clinical variables were treated as secondary endpoints.

4.      Results

4.1.Table 1  should include  BMI, proportion of  obese participants and diabetes duration

Authors' Answer

Following the recommendations of the reviewer, we have included BMI, proportion of obese participants and diabetes duration in Table 1.

4.2.a) Figure 2 the analysis reported in this figure should be better described in the text. The meaning of “suitable adherence to the Mediterranean diet and quality of diet” should be clarified;

Authors' Answer

The meaning “suitable adherence to the Mediterranean diet and the quality of diet” can be observed in the methodology section (Page 5 Line 182 and Page 6 Line 192). However, following the recommendations of the reviewer, we have described the analysis of Figure 2, and we have added a clarification on these terms in the results section

Page 10 Line 275: Figure 2 shows the evolution of the percentage of compliance of the Mediterranean diet and the DQI index in the IG and CG. At the follow-up visits at 3 and 12 months, the number of subjects with suitable adherence to the Mediterranean diet (MEDAS score 9 points) in the IG increased significantly compared to the CG. In the same way, there was an increase in the IG of the number of subjects with a suitable quality of diet (DQI score 40 points), with respect to the CG. This difference was significant in the follow-up visit at 3 months. (p<0.05 for all).

b) Furthermore these measures  were not described  among the study end points

Authors' Answer

As we have previously commented, the endpoint of this study is the total score of the MEDAS questionnaire. Figure 2 describes a complementary information that we have introduced to improve the understanding of the article, but it is not an endpoint.

4.3.Figure 3 are these numbers the same reported in table 2 (first two lines)? If  not, please explain the difference? If yes, please explain why a statistical significance is observed in the figure but not in the table.

Authors' Answer

Figure 3 and Table 2 do not report the same data. Table 2 shows the means, along with the standard deviations, of the values that take the MEDAS questionnaire and the DQI questionnaire throughout the study, this table is simply descriptive. However, Table 3 shows the differences that have occurred in follow-up visits at 3 and 12 months with respect to the baseline visit. Finally, Figure 3 shows the analysis performed through repeated measures ANOVA test with adjustment for baseline visit, which shows the evolution of the scores throughout the study.

We have considered removing the phrase "*There are significant differences between the control and intervention group at the baseline visit" from Table 2 and Table 3 as we think it may be misleading.

In addition, we have modified the order of Table 2, Table 3 and Figure 3 to improve the understanding of the manuscript.

4.4. a) Data on clinical variables are relevant as the final aim of improving  adherence to diet is to improve the management of diabetes, they should be  given in the paper  and not reported as appendix tables.

Authors' Answer

Following the recommendations of the reviewer, we have included the appendix tables within the manuscript (Table 4 and Table 5).

Page 12 Line 292: Finally, without significant changes in the consumption of antidiabetic drugs, the following improvements were observed at the 3-month follow-up visit compared to the baseline visit: a postprandial glucose improvement of -9.4 mg/dl (-18.0 – -0.8), a waist circumference improvement of -2.5 cm (-3.3 – -1.6) and a BMI improvement of -0.3 kg/m2 (-0.6 – 0.0) in favour of the IG (p<0.05 for all; Table 4 and Table 5). The improvements of some of the clinical parameters were maintained at the follow-up visit at 12 months compared to baseline, such as a postprandial glucose improvement of -6.6 mg/dl (-14.9 – 1.7), waist circumference improvement of -0.9 cm (-3.1 – 1.2) and systolic blood pressure improvement of -1.5 mmHg (-8.7 – 5.7) However, the results did not reach statistical significance (p>0.05).

b)     With regard to these data it is important to clarify whether changes in the use of drugs occurred  between baseline and follow-up.

Authors' Answer

After having verified that there were no significant changes in the antidiabetic treatment, we modified the following sentence:

Page 12 Line 292: Finally, without significant changes in the consumption of antidiabetic drugs, the following improvements were observed at the 3-month follow-up visit compared to the baseline visit: a postprandial glucose improvement of -9.4 mg/dl (-18.0 – -0.8), a waist circumference improvement of -2.5 cm (-3.3 – -1.6) and a BMI improvement of -0.3 kg/m2 (-0.6 – 0.0) in favour of the IG (p<0.05 for all; Table 4 and Table 5). The improvements of some of the clinical parameters were maintained at the follow-up visit at 12 months compared to baseline, such as a postprandial glucose improvement of -6.6 mg/dl (-14.9 – 1.7), waist circumference improvement of -0.9 cm (-3.1 – 1.2) and systolic blood pressure improvement of -1.5 mmHg (-8.7 – 5.7) However, the results did not reach statistical significance (p>0.05).

Reviewer 2 Report

I carefully read the manuscript by Alonso-Domínguez and collaborators, which is overall interesting. Some comments in order to improve the manuscript: 

- I think that the authors should better characterized the intervention, because the description is actually confused and dispersive. For example, I think that the information in the paragraph 2.3.1 could be included in the paragraph 2.2 and so on, in order to improve the readability of the methods section.

 - Line 209: Fisher exact test is also supposed to be performed.

 - Authors stated that they collected much more parameters that the ones reported in the manuscript and in the supplementary material (i.e. waist circumference, total serum cholesterol and HDL-cholesterol). Please, include also these parameters in the tables 1S and 2S.

 - Authors are supposed to check the English language throughout the manuscript, in order to correct the typos errors.

Author Response

REVIEWER #2:

I carefully read the manuscript by Alonso-Domínguez and collaborators, which is overall interesting. Some comments in order to improve the manuscript:

Authors' Answer

First, thank you for your work in reviewing this manuscript. Your contributions and suggestions will improve the understanding of the text.

1.      I think that the authors should better characterized the intervention, because the description is actually confused and dispersive. For example, I think that the information in the paragraph 2.3.1 could be included in the paragraph 2.2 and so on, in order to improve the readability of the methods section.

Authors' Answer

Following the recommendations of the reviewer, we modified the material and methods section, with the aim of improving its understanding, remaining as follows:

Page 2 Line 85:

2. Materials and Methods

2.1. Study design

The EMID study [22] (Effectiveness of a multifactorial intervention based on an application for smartphones, a nutritional workshop, and heart-healthy walks, in patients with T2DM in primary care) is a randomized, controlled clinical trial with two parallel groups and a follow-up period of 12 months. The study was carried out in the field of primary health care, in the Alamedilla Research Unit, which belongs to the Research Network on Preventive Activities and Health Promotion (redIAPP) and the Biomedical Research Unit of Salamanca (IBSAL).

2.2. Participants

One of the objectives was to obtain a sample that is the most representative of the studied population. Therefore, the subjects were selected by stratified random sampling from among the patients with T2DM who sought medical attention at the Alamedilla Health Centre. The subjects were divided according to their age into two groups (25-50 and 51-70 years).

The inclusion criteria were age between 25 and 70 years, T2DM, agreement to participate in the study, and signing an informed consent document after receiving information about the study. As diagnostic criteria for T2DM, we followed the latest recommendations of the American Diabetes Association: fasting plasma glucose above 126 mg/dL or two-hour plasma glucose above 200 mg/dL during an oral glucose tolerance test (using a glucose load containing the equivalent of 75 g anhydrous glucose dissolved in water) or glycosylated haemoglobin over 6.5%. In all cases, these tests were repeated to confirm the results in the absence of unequivocal hyperglycaemia. Additionally, patients were also considered as having T2DM if they had the classic symptoms of hyperglycaemia or hyperglycaemic crisis (i.e. random plasma glucose above 200 mg/dL) [23].

The exclusion criteria were a history of cardiovascular events, musculoskeletal pathology that prevents walking, and clinically demonstrable neurological or neuropsychological disease, which would prevent the subject from visiting the health centre.

2.3. Common advice

All participants received standardized counselling for 10 minutes about healthy eating and physical activity. The food section lasted five minutes and focused on the use of the plate method and recommendations to help comply with the Mediterranean diet. Five minutes were also used to give advice to help comply with current international recommendations regarding physical activity. A brochure was given to the participants for support in both areas.

2.4. Randomisation and masking

Participants were randomized at 1:1 into the control group (CG) (102) or the intervention group (IG) (102). The randomization was performed after obtaining informed consent and was not revealed prior to group assigment. The allocation sequence was generated by an independent researcher using the software Epidat 4.0 (Figure 1).

Due to the nature of the intervention, the participants could not be blinded. However, the researcher who carried out the intervention in the study group was different from the person responsible for the assessment and standardized counselling. In addition, the person responsible for the statistical analysis remained blinded throughout the study. During the follow-up visits, patients were told that they should not use other health technologies. Moreover, the application was not made available online until the end of the study so that the control group could not access it.

2.5. Intervention

A multifactorial intervention was carried out with groups of 10 participants consisting of a food workshop, a smartphone application and heart-healthy walks. To impart this multifactorial intervention, three nurses from the health centre were instructed in two one-hour training classes on how to carry out the interventions in a standardized manner (what points should be treated, in what order, and for how long).

2.5.1. Food workshop

The food workshop was a theoretical and practical workshop that lasted 90 minutes and focused on improving adherence to the Mediterranean diet. The workshop covered the following topics: benefits of a healthy diet, food groups, components of the Mediterranean diet, recommended culinary techniques, the use of the plate method and the importance of food labelling for patients with diabetes. Adherence to the food workshop was assessed by the attendance or lack thereof.

2.5.2. Smartphone application (EVIDENT II)

A one-hour group workshop was held to instruct the participants in the use of the EVIDENT II application (intellectual property registry number SA-81-14), which was installed on a mobile phone that was provided for them to use for three months. This application was designed by software engineers in collaboration with dieticians and physical activity experts with the aim of promoting adherence to the Mediterranean diet and it has already been used in previous studies [24,25]. The application was configured with the data of each participant (age, sex, weight, height, and stride distance). After entering food intake and daily exercise data, it would provide detailed information on nutritional deviations in terms of both diet composition and the number of calories, with the aim of encouraging a change of habits. The cell phone was returned after three months, at follow-up visit for both groups. Subsequently, the stored information was downloaded. Adherence to the smartphone app was assessed by the number of days of recordings in the device. After this three-month intervention period, the subjects did not have access to the EVIDENT II app because it was not freely available online.

2.5.3. Heart-healthy walks

Once a week for 5 weeks, the subjects performed 10 minutes of warm-up, walked 4 km on flat terrain, and performed 10 minutes of stretching and relaxation. In order to make the walks qualify as aerobic exercise (50 - 70% maximum heart rate) [26], participants were divided into two groups according to on intensity. The approximate speed was 6 km/hour for the group walking at moderate intensity (5 metabolic equivalents [METs]) and 3 - 4 km/hour for the group walking at low intensity (2.5 METs). Adherence to heart-healthy walks was evaluated by the number of days, planned in the intervention that the subjects attended to perform them.

2.6. Outcomes measures and follow-up

To assess the effect of the multifactorial intervention, follow-up was carried out at baseline, three months, and 12 months after the initial intervention. As the main endpoint, we considered the change in total score of the Mediterranean Diet Adherence Screener (MEDAS) questionnaire [27], while the total score of the Diet Quality Index (DQI) questionnaire [28] and the clinical variables were treated as secondary endpoints.

2.6.1. Adherence to the Mediterranean diet

The main result variable was measured using the validated fourteen-item MEDAS questionnaire, developed by the PREDIMED group, which includes 12 questions about the frequency of food consumption and two questions about typical eating habits for the Spanish population [27]. Each question was scored with zero or one point. One point was given for using olive oil as the main fat for cooking, preferring white meat to red meat, and daily consumption of four or more tablespoons (one tablespoon = 13.5 g) of olive oil (including oil used for frying, dressing salads, etc.), two or more servings of vegetables, three or more pieces of fruit, less than one serving of red meat or sausage, less than one serving of animal fat, and less than one cup (one cup = 100 ml) of carbonated or sugary drinks. One point was also given for weekly intake of seven or more glasses of wine, three or more servings of legumes, three or more servings of fish, two shop-bought pastries or fewer, three or more servings of nuts, and two or more helpings of sofrito (a traditional sauce made with tomato, garlic, onion, or leeks, and sautéed with olive oil). The final score range was 0 to 14 points, with 9 or more points indicating suitable adherence to the Mediterranean diet [27].

2.6.2. Diet Quality Index

The diet was assessed with the DQI [28]. This questionnaire covers 18 food groups, labelled one, two or three depending on whether their consumption is beneficial (eating more results in a higher score) or harmful to health (eating more produces a lower score). Food groups are classified into three categories according to the recommended frequency of use. The first category includes eight foods, and their response types are "less than once a day", "once a day", or "more than once a day". The second category has seven foods and their responses are "less than four times a week", "four to six times a week", or "at least once a day". Finally, in the third category, there are three foods, which are classified as "less than twice a week", "two to three times a week", or "four or more times a week" [28]. Scores range from 18 to 54 points, with a higher score representing higher diet quality, and 40 or more points indicate a suitable quality of diet.

2.6.3. Clinically relevant measures

Other variables were measured, including drug use, blood pressure, postprandial glucose, weight, height, waist circumference (WC), and biochemical parameters (total serum cholesterol, LDL-cholesterol, HDL-cholesterol). Body mass index (BMI) was calculated by dividing the weight (kg) by the square of the height (m2). The body shape index (ABSI) was estimated with the following equation: ABSI= WC (m)/[BMI2/3*height (m)1/2]. A detailed description of the way in which these variables were measured was published with the study protocol [22].

2.7. Sample size calculation

The sample size was estimated a priori, considering the increase in the total score of the MEDAS questionnaire as the main endpoint. Assuming alpha=0.05 and beta=0.20 with an SD of 2 points, we needed 140 participants (70 per group) in order to detect an increase of 1 point in the IG's MEDAS score compared to the CG while allowing for an expected drop-out rate of 10%. In addition, for an SD of 2.4 points in the DQI score, we needed 202 participants (101 per group) in order to detect an increase of 1 point in the IG’s DQI score compared to the CG with the same dropout rate. Therefore, we considered 204 subjects to be a sufficient number for detecting clinically relevant differences in the main variables of the study.

2.8. Ethical considerations

The study was approved by the Clinical Research Ethics Committee of the Health Area of Salamanca on November 28, 2016. All procedures were performed in accordance with the ethical standards of the institutional research committee and with the 2013 Declaration of Helsinki [29]. All patients signed written informed consent documents prior to participation in the study. The trial was registered at ClinicalTrials.gov with identifier NCT02991079.

2.9.Statistical analysis

Continuous variables are presented as means ± standard deviations and qualitative variables are presented using a frequency distribution. To compare categorical variables at baseline between IG and CG the chi-squared test or Fisher exact test was carried out as appropriate, and quantitative variables were compared using the student’s t-test. The ANCOVA test was used to compare the changes between the IG and the CG, adjusting for the basal measurement of each variable. Repeated-measures analysis of variance was used to analyse group interaction effects on changes in the Mediterranean diet and DQI scores using the GLM procedure. For the bilateral contrast of hypotheses, an alpha risk of 0.05 was set as a limit of statistical significance. The data were analysed using the statistical software SPSS for Windows version 25.0. (IBM, Armonk, New York: IBM Corp).

2.      Line 209: Fisher exact test is also supposed to be performed.

Authors' Answer

In order to improve the understanding of the manuscript, we have added this test in the statistical analysis section:

Page 6 Line 218: To compare categorical variables at baseline between IG and CG the chi-squared test or Fisher exact test was carried out as appropriate, and quantitative variables were compared using the student’s t-test.

3.      Authors stated that they collected much more parameters that the ones reported in the manuscript and in the supplementary material (i.e. waist circumference, total serum cholesterol and HDL-cholesterol). Please, include also these parameters in the tables 1S and 2S.

Authors' Answer

Following the recommendations of the reviewer we have included waist circumference, total serum cholesterol and HDL-cholesterol in the Table 4 (Table 1S) and Table 5 (Table 2S). At the request of another review, we have included the appendix tables (Table 1S and Table 2S) within the manuscript.

We have also included this information in the results section:

Page 12 Line 292: Finally, without significant changes in the consumption of antidiabetic drugs, the following improvements were observed at the 3-month follow-up visit compared to the baseline visit: a postprandial glucose improvement of -9.4 mg/dl (-18.0 – -0.8), a waist circumference improvement of -2.5 cm (-3.3 – -1.6) and a BMI improvement of -0.3 kg/m2 (-0.6 – 0.0) in favour of the IG (p<0.05 for all; Table 4 and Table 5). The improvements of some of the clinical parameters were maintained at the follow-up visit at 12 months compared to baseline, such as a postprandial glucose improvement of -6.6 mg/dl (-14.9 – 1.7), waist circumference improvement of -0.9 cm (-3.1 – 1.2) and systolic blood pressure improvement of -1.5 mmHg (-8.7 – 5.7) However, the results did not reach statistical significance (p>0.05).

4.      Authors are supposed to check the English language throughout the manuscript, in order to correct the typos errors.

Authors' Answer

Following the recommendations of the reviewer, we have sent to edit the manuscript to American Manuscript Editors. We attach the edition certificate.

Round 2

Reviewer 1 Report

I am satisfied with the revision. A few points  still need attention in my opinion

1)Page 16 Line 350: I would suggest the following change: The results of this study suggest that the proposed multifactorial intervention involving a food workshop and a smartphone application are moderately effective in  improving adherence to the Mediterranean diet and diet quality among patients with T2DM.

 2) Considering that  the  calculation  of the  sample size was made assuming as the outcome measure   an increase of 1 point in the IG's MEDAS score, the proportion of participants meeting this end point in the  intervention and control group should be given somewhere in the results section

table 3 the statistical significance for the comparisons should  be provided

Author Response

REVIEWER #1:

General comments

I am satisfied with the revision. A few points still need attention in my opinion

Authors' Answer

First, thank you for your work in reviewing this manuscript. Your contributions and suggestions will improve the understanding of the text.

Specific commnets

1)      Page 16 Line 350: I would suggest the following change: The results of this study suggest that the proposed multifactorial intervention involving a food workshop and a smartphone application are moderately effective in  improving adherence to the Mediterranean diet and diet quality among patients with T2DM.

Authors' Answer

Following the recommendations of the reviewer, we have modified the conclusions, remaining as follows:

Page 16 Line 355: The results of this study suggest that the proposed multifactorial intervention involving a food workshop and a smartphone application are moderately effective in improving adherence to the Mediterranean diet and diet quality among patients with T2DM.

2)      Considering that  the  calculation  of the  sample size was made assuming as the outcome measure an increase of 1 point in the IG's MEDAS score, the proportion of participants meeting this end point in the intervention and control group should be given somewhere in the results section.

Authors' Answer

Taking into account the recommendation of the reviewer, we have included the following paragraph in the results section:

Page 10 Line 263: At the 3-month follow-up visit, 82.7% of the IG participants increased their MEDAS questionnaire score by at least 1 point, while in the CG this value was 31.3 %; at the 12-month follow-up visit, 61.9% of the IG participants and 38.5% of the CG participants reached this increase.

3)      Table 3 the statistical significance for the comparisons should  be provided

Authors' Answer

We have included the statistical significance in the foot of table of Table 3 and Table 5.

Page 10 Line 270: Significant difference: p < 0.05

Page 15 Line 312: Significant difference: p < 0.05
